# Early IL-17A production helps establish *Mycobacterium intracellulare* infection in mice

**Bock-Gie Jung** [1]*, **Buka Samten**[1], **Kristin Dean**[1], **Richard J. Wallace, Jr.**[2], **Barbara A. Brown-Elliott**[2], **Torry Tucker** [3], **Steven Idell** [3,4], **Julie V. Philley**[5], **Ramakrishna Vankayalapati**[1]

1 Department of Pulmonary Immunology, The University of Texas Health Science Center at Tyler, Tyler, Texas, United States of America, 2 Department of Microbiology, The University of Texas Health Science Center at Tyler, Tyler, Texas, United States of America, 3 Department of Cellular and Molecular Biology, The University of Texas Health Science Center at Tyler, Tyler, Texas, United States of America, 4 The Texas Lung Injury Institute, Tyler, Texas, United States of America, 5 Department of Medicine, The University of Texas Health Science Center at Tyler, Tyler, Texas, United States of America

* bockgie.jung@uthct.edu

**Data Availability Statement:** All relevant data are within the manuscript and its Supporting information files.

## Abstract

Nontuberculous mycobacteria (NTM) infection is common in patients with structural lung damage. To address how NTM infection is established and causes lung damage, we established an NTM mouse model by intranasal inoculation of clinical isolates of *M. intracellulare*. During the 39-week course of infection, the bacteria persistently grew in the lung and caused progressive granulomatous and fibrotic lung damage with mortality exceeding 50%. Lung neutrophils were significantly increased at 1 week postinfection, reduced at 2 weeks postinfection and increased again at 39 weeks postinfection. IL-17A was increased in the lungs at 1–2 weeks of infection and reduced at 3 weeks postinfection. Depletion of neutrophils during early (0–2 weeks) and late (32–34 weeks) infection had no effect on mortality or lung damage in chronically infected mice. However, neutralization of IL-17A during early infection significantly reduced bacterial burden, fibrotic lung damage, and mortality in chronically infected mice. Since it is known that IL-17A regulates matrix metalloproteinases (MMPs) and that MMPs contribute to the pathogenesis of pulmonary fibrosis, we determined the levels of MMPs in the lungs of *M. intracellulare*-infected mice. Interestingly, MMP-3 was significantly reduced by anti-IL-17A neutralizing antibody. Moreover, *in vitro* data showed that exogenous IL-17A exaggerated the production of MMP-3 by lung epithelial cells upon *M. intracellulare* infection. Collectively, our findings suggest that early IL-17A production precedes and promotes organized pulmonary *M. intracellulare* infection in mice, at least in part through MMP-3 production.

## Author summary

To determine how nontuberculous mycobacteria (NTM) infection is established and how NTM disease progresses, we established a chronic NTM mouse model by intranasal inoculation of *M. intracellulare*, one of the most frequently isolated strains in NTM patients.

**Funding:** The author(s) received no specific funding for this work.

**Competing interests:** The authors have declared that no competing interests exist.

The bacteria persistently grew in the lungs and caused fibrotic lung damage with over 50% mortality over 39 weeks. Neutrophils and IL-17A rapidly increased in the lung during early (1–2 weeks) infection, and neutrophils reappeared at 39 weeks postinfection. Depletion of neutrophils during early (0–2 weeks) and chronic (32–34 weeks) infection had no effect on mortality or lung damage in chronically infected mice. Neutralization of IL-17A during early (0–2 weeks) infection significantly reduced mortality, bacterial burden, fibrotic lung damage, and lung matrix metalloproteinase (MMP)-3 at 39 weeks postinfection. Exogenous IL-17A exaggerated the production of MMP-3, but not MMP-9, by lung epithelial cells upon *M. intracellulare* infection. This study demonstrates that early IL-17A production contributes to established *M. intracellulare* infection in mice.

## Introduction

Nontuberculous mycobacteria (NTM) include all species belonging to the genus *Mycobacterium* except *M. tuberculosis* and *M. leprae*, which cause tuberculosis (TB) and leprosy, respectively [1]. The *M. avium* complex is the most frequently isolated group in NTM patients, and it mainly consists of *M. avium* and *M. intracellulare* [2–4]. During the past two decades, the incidence of disease caused by NTM [2–6] and its economic burden have increased worldwide [7,8]. There are 2 major types of NTM infection. Pulmonary NTM infections are the most common and mainly affect elderly postmenopausal women or individuals with pre-existing lung diseases [9]. Disseminated NTM infections may occur in immunocompromised individuals [9], but they may also occur in apparently immunocompetent individuals [1,10].

High levels of inflammatory cytokines and increased leukocyte numbers were observed in the bronchoalveolar lavage fluid (BALF) of NTM patients compared with healthy subjects [11,12]. Levels of total protein, albumin, and lactate dehydrogenase were significantly higher in the BALF of NTM patients than in healthy subjects [11]. These clinical observations suggest that NTM infection causes inflammatory lung injury. Indeed, NTM infection is common in patients with structural lung damage, including bronchiectasis, chronic obstructive lung disease (COPD), and cystic fibrosis (CF) [1,9]. However, it remains unclear how NTM infection is established and causes lung damage.

Interleukin (IL)-17A and IL-17A-mediated immune responses are also related to lung damage caused by aberrant inflammation in various pulmonary diseases, such as COPD [13] and CF [14]. IL-17A exaggerates inflammatory responses in respiratory epithelial cells [15] and promotes the expression of α-smooth muscle actin and the production of profibrotic factors in fibrocytes [16]. IL-17A is produced by various immune cell types, including γδ T cells, invariant natural killer T (iNKT) cells, and type 3 innate lymphoid cells (ILC3s) [17,18]. A previous study showed that NTM induce IL-17A production by human T cells [19]. The receptor for IL-17A is a heteromeric complex composed of the IL-17 receptor (IL-17R)A and IL-17RC, and it is expressed on the surface of various cell types, including epithelial cells, fibroblasts, keratinocytes, neutrophils, and eosinophils [18,20]. Ligation of IL-17A to the receptor complex (IL-17RA/RC) induces ACT1-TRAF6-TAK1-mediated activation of the NF-κB and MAPK pathways, which leads to the production of proinflammatory cytokines and chemokines [20]. Serum IL-17A levels are significantly increased in NTM patients compared with healthy subjects and can be reduced by antibiotic treatment [21]. However, impaired IL-17A responses have also been observed in NTM patients [22,23]. Thus, the role of IL-17A in the immunopathogenesis of NTM remains controversial and unclear.

Matrix metalloproteinases (MMPs) are a family of zinc-dependent endopeptidases, and some MMPs are involved in fibrosis of various organs [24–27]. In particular, MMP-3 has been directly implicated in epithelial-mesenchymal transformation (EMT) and in the pathogenesis of pulmonary fibrosis [28,29]. A previous study showed that MMP-3 production was increased in human small airway epithelial cells (SAECs) and normal human bronchial epithelial cells (NHBEs) in the presence of exogenous IL-17A. These findings suggest that MMP-3 may be involved in IL-17A-mediated pulmonary fibrosis.

In the present study, we established an NTM mouse model by intranasal inoculation of clinical isolates of *M. intracellulare*. We then used the model to explore how NTM infection is established and causes lung damage. We demonstrated that excessive IL-17A production during the first 2 weeks of infection leads to increased bacterial growth, fibrotic lung damage, and mortality in the chronic phase of infection. MMP-3 production contributes to these effects. These imbalances predispose mice to progressive granulomatous organization with pulmonary fibrosis and increased mortality during prolonged infection.

## Results

### *M. intracellulare* growth in mice

To evaluate bacterial growth, mice were infected with clinical isolates of *M. intracellulare* via the intranasal route. A schematic representation of the bacterial infection and experimental schedule is shown in Fig 1A. The bacterial burden was determined in the lungs, spleen, mediastinal lymph nodes (MLNs), and liver at 1, 2, 3, 4, 8, 12 and 39 weeks postinfection (Fig 1B–1E). The lung bacterial burden was $7.23 \pm 0.06$ log colony-forming units (CFU) at 1 week postinfection. It continuously increased to $7.78 \pm 0.52$ log CFU ($P < 0.001$) at 4 weeks postinfection, $8.07 \pm 0.11$ log CFU ($P < 0.001$) at 12 weeks postinfection and $8.49 \pm 0.21$ $\log_{10}$ CFU ($P < 0.001$) at 39 weeks postinfection (Fig 1B). Bacteria were also detected in extrapulmonary organs within 1 week of infection and grew continuously during the 39-week course of infection (Fig 1C–1E).

### Mortality of mice following *M. intracellulare* infection

We performed a separate mortality study. Twenty mice per group were dedicated to the mortality study and experiments were repeated two times. Thus, mortality was analyzed from a total of 40 mice. All uninfected control mice survived and 75% of infected mice died during the 39 weeks of infection (Fig 1F, $P < 0.001$). Although 10% of the infected mice died within 8 weeks of infection, the majority of deaths occurred after 20 weeks of infection. We also measured the body weights of infected mice until 39 weeks postinfection. As shown in S4A Fig, infected mice demonstrated significant weight loss within 2–3 weeks of infection, after which they partially regained and then lost weight at the time of death.

### Lung pathological changes in mice infected with *M. intracellulare*

Pathological changes were next evaluated in the lungs of *M. intracellulare*-infected mice at 1, 2, 3, 4, 8, 12 and 39 weeks postinfection. The lungs were sectioned and stained with hematoxylin and eosin (Fig 2A) and trichrome (Fig 2B). Perivascular and peribronchial cuffing was observed throughout the lung and alveolar inflammation was observed as early as 1–2 weeks following infection. These inflammatory lesions gradually increased, and granulomatous-like and fibrotic lesions were observed in the lung at 8–12 weeks postinfection. Subsequently, typical granulomas surrounded by necrotic and fibrotic lesions were found in the lungs at 39

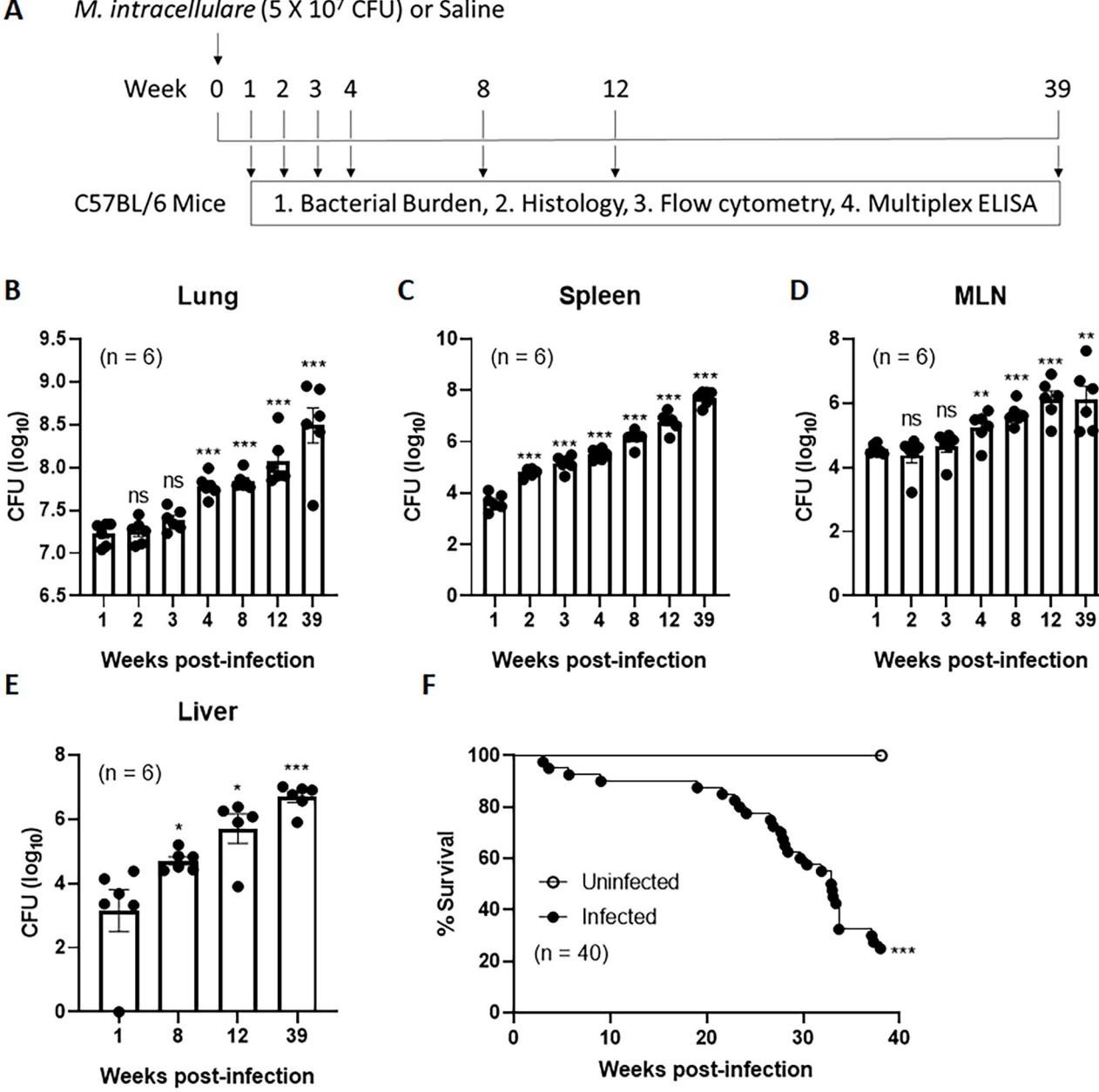

**Fig 1. *M. intracellulare* infection causes persistent bacterial growth and mortality in mice.** (**A**) Schematic representation of *M. intracellulare* infection and experimental schedule. C57BL/6 mice were infected with clinical isolates of *M. intracellulare* ($5 \times 10^7$ CFU) via the intranasal route. The bacterial burden in the lungs after 24 h was $3.3 \times 10^7$ CFU. Mice were sacrificed at 1, 2, 3, 4, 8, 12 and 39 weeks postinfection. Bacterial burden, histological analyses, immune cell subpopulation (by flow cytometry) and cytokine/chemokine levels (by multiplex ELISA) were evaluated at each indicated time. (**B-E**) Bacterial burden was determined in the lungs (B), spleen (C), mediastinal lymph nodes (MLN) (D), and liver (E). Data were pooled from two independent experiments (total n = 6 mice per indicated time point). Data are expressed as the means ± SEM. *P < 0.05, **P < 0.01, and ***P < 0.001 compared with 1 week postinfection. (**F**) A mortality study was performed separately. Twenty mice per group were used in this study and experiments were repeated two times. Survival curves for uninfected control (white circles) and *M. intracellulare*-infected mice (black circles). Data were pooled from two independent experiments (total n = 40 mice per group). Survival curves were compared using the log rank test. ***P < 0.001.

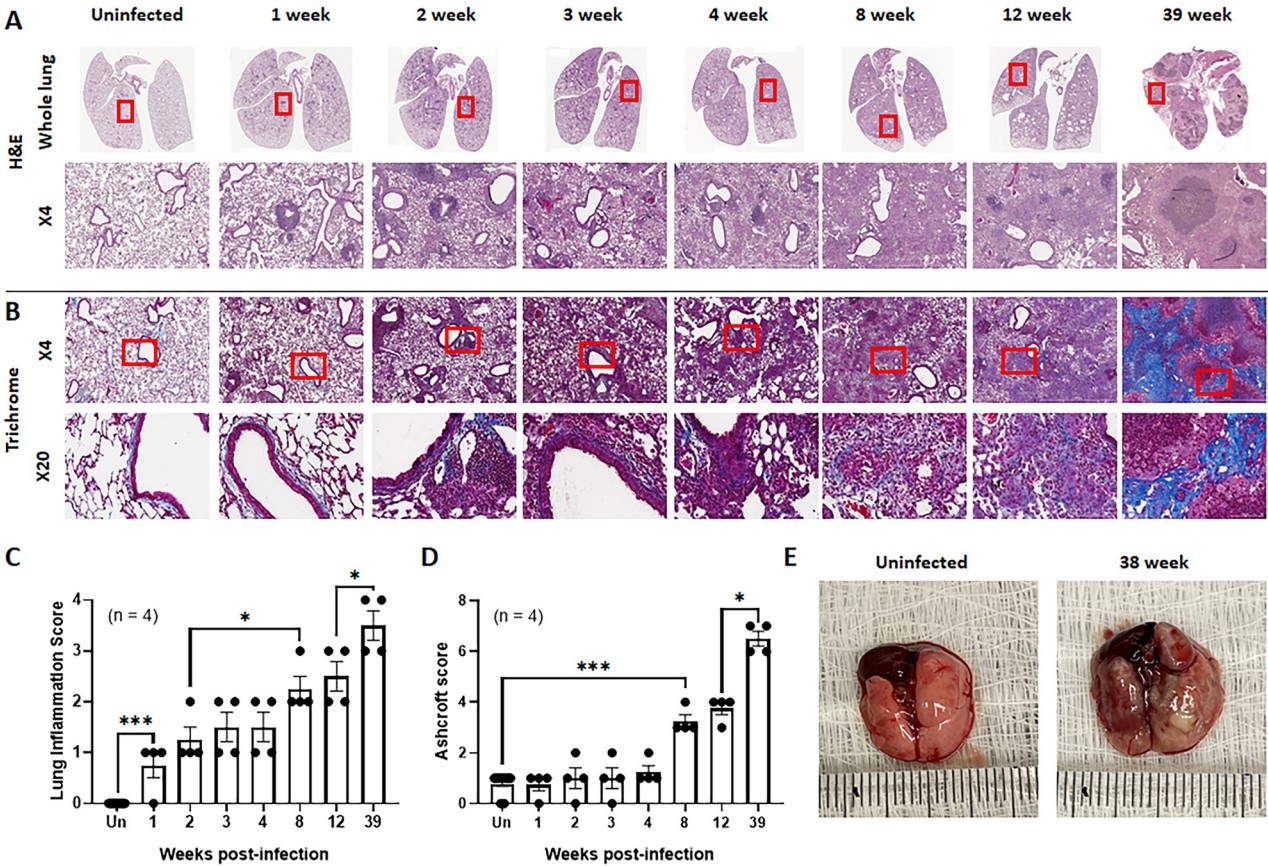

**Fig 2. *M. intracellulare* infection induces lung inflammation and fibrosis in mice.** (**A-E**) C57BL/6 mice were infected with clinical isolates of *M. intracellulare* ($5 \times 10^7$ CFU) via the intranasal route. Mice were sacrificed at 1, 2, 3, 4, 8, 12 and 39 weeks postinfection. Lungs from uninfected control and *M. intracellulare*-infected mice were collected and formalin-fixed. Paraffin-embedded tissue sections were prepared (total n = 4 mice per indicated time point) and stained with hematoxylin and eosin (A) and trichrome (B). A representative figure of each indicated time point is shown (A, B). (C) The severity of lung inflammation was quantified from a total of 4 mice per indicated time point using a score from 0 (no inflammation) to 4 (severe inflammation) for each of the following criteria: alveolar wall inflammation, alveolar destruction, leukocyte infiltration, and perivascular inflammation. (D) The severity of pulmonary fibrosis was quantified from a total of 4 mice per indicated time point according to the Ashcroft scoring system. All histopathological evaluations were performed in a blinded fashion. (E) A representative lung gross morphology of uninfected control (left) and *M. intracellulare*-infected mice (right) at 38 weeks postinfection is shown. Data are expressed as the means ± SEM. $^*P < 0.05$ and $^{***}P < 0.001$.

weeks postinfection. The lung inflammatory score (Fig 2C) and Ashcroft score (Fig 2D) gradually increased over time. *M. intracellulare*-infected mice exhibited gross lung lesions at 38 weeks postinfection (Fig 2E).

## Immune cell infiltration in the lungs of mice infected with *M. intracellulare*

Immune cell populations were identified and evaluated in the lungs of *M. intracellulare*-infected mice at 1, 2, 3, 4, 8, 12 and 39 weeks postinfection. Total lung CD45+ cells were significantly increased in infected mice compared with uninfected mice at 1 week postinfection ($2.60 \times 10^6 \pm 0.37$ vs. $1.20 \times 10^6 \pm 0.06$ cells, $P < 0.001$). These cells were subsequently increased at 2 weeks postinfection ($7.00 \times 10^6 \pm 0.35$ cells, $P < 0.001$). A gradual decline in lung CD45+ cells was observed 4 weeks postinfection, with comparable levels at 39 weeks postinfection ($1.41 \times 10^6 \pm 0.82$ cells) in infected versus uninfected control mice (Fig 3A). Similar trends were noted with interstitial macrophages (IMs), alveolar macrophages (AMs), dendritic

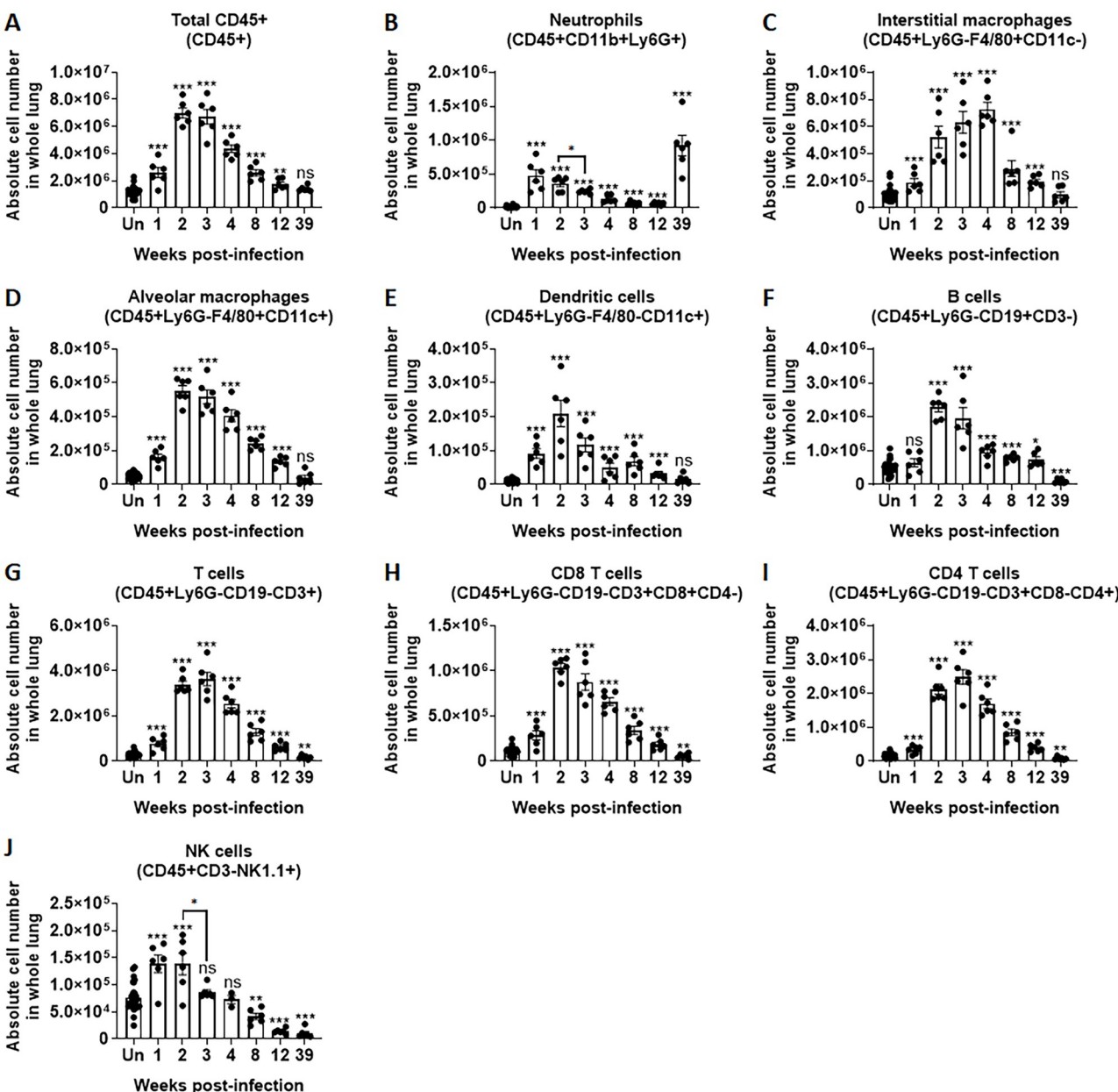

**Fig 3. Lung infiltration of immune cells in *M. intracellulare*-infected mice.** (A-J) Lungs from uninfected control and *M. intracellulare*-infected mice were collected at 1, 2, 3, 4, 8, 12 and 39 weeks postinfection. The absolute number of immune cells per whole lung was determined by flow cytometry. Data were pooled from two independent experiments (uninfected mice n = 30, infected mice n = 6 mice per indicated time point). Data are expressed as the means ± SEM. *P < 0.05, **P < 0.01, and ***P < 0.001. ns, not significant. P values for comparison between uninfected mice (Un) and other were indicated with asterisks sans connecting lines. For other comparisons, connecting lines used.

cells (DCs), B cells, T cells, CD8 T cells, CD4 T cells and NK cells (Fig 3C–3J). Of note, a unique double peak of neutrophilic pulmonary infiltration was noted. Neutrophils were significantly increased at 1 week postinfection, reduced at 2 weeks and increased again at 39 weeks postinfection (Fig 3B).

## Cytokine and chemokine responses in the lungs of mice infected with *M. intracellulare*

We next sought to determine whether *M. intracellulare* infection had any effects on pro- and anti-inflammatory responses. Cytokine and chemokine levels were measured in lung homogenates at 1, 2, 3, 4, 8, 12 and 39 weeks postinfection by multiplex (36-plex) ELISA (Fig 4 and S2 Fig). Most of the cytokines and chemokines were elevated and reached a peak at 2–3 weeks postinfection and gradually declined over time (Fig 4A and S2 Fig). Conversely, Gro-α (KC), G-CSF, MIP-1α, MIP-2 and IL-10 reappeared and were significantly increased at 39 weeks postinfection (S2 Fig). Of note, IL-17A rapidly increased, reached a peak at 1–2 weeks postinfection and then declined at 3 weeks postinfection (Fig 4B). Hierarchical clustering was performed on cytokine and chemokine profiles obtained from 57 samples [uninfected mice (n = 15) and infected mice (n = 6 mice/each time point)]. Based on visual inspection of the heatmap and dendrogram, the expression profile of IL-17A was dissimilar to that of any other cytokines or chemokines tested by multiplex ELISA (Fig 4A). These results suggested that IL-17A has unique expression kinetics compared to other cytokines and chemokines in mice infected with *M. intracellulare*. The results also suggest that it may be involved in early *M. intracellulare* infection and mainly produced by CD3+ cells, especially RORγt+ cells (Fig 4C and S3 Fig).

## Early and late neutrophil infiltration of the lungs has no effect on mortality or lung damage in chronically infected mice

Excess neutrophil infiltration of the lungs is associated with inflammation, tissue damage and even death in some cases [30]. In our NTM model, neutrophils were significantly increased at 1 week (Fig 3B) and reduced after 2 weeks of infection. In contrast to the other myeloid cells, neutrophil reappearance at 39 weeks postinfection coincided with 75% mortality, suggesting that pulmonary infiltration of neutrophils may contribute to mortality of the mice. Thus, we tested the contribution of early (during 0–2 weeks of infection) or late neutrophil (during 32–34 weeks of infection) infiltration of the lungs to mortality following infection with *M. intracellulare*. Fig 5A and 5I illustrate the schematic representation of the interventional approach using anti-Ly-6G monoclonal antibody (mAb) or isotype-matched control Ab treatment to deplete neutrophils. As shown in S5A Fig, neutrophils were completely depleted in the lungs of mice treated with anti-Ly-6G mAb.

Depletion of neutrophils at early infection did not affect the mortality of the mice until 48 weeks postinfection (Fig 5B). Lung computed tomography scan images showed comparable parenchymal lung infiltrates in *M. intracellulare*-infected mice (both anti-Ly-6G mAb- and isotype-matched control Ab-treated mice) compared with uninfected control mice at 48 weeks postinfection (Fig 5C). Indeed, lung volume was significantly reduced in *M. intracellulare*-infected mice compared with uninfected control mice (Fig 5E). However, no difference in the lung volume of anti-Ly-6G mAb- or isotype-matched control Ab-treated mice was noted. Moreover, elastance and resistance were significantly increased and lung compliance was reduced in the lungs of *M. intracellulare*-infected mice (both anti-Ly-6G mAb- and isotype-matched control Ab-treated mice) compared with uninfected control mice at 48 weeks postinfection (Fig 5F–5H). No differences in the elastance, compliance or resistance of the lungs of anti-Ly-6G mAb- and isotype-matched control Ab-treated mice were noted. These findings suggested that early lung neutrophil infiltration does not play a significant role in mortality or lung restriction in *M. intracellulare* pulmonary infection.

Depletion of neutrophils at a later stage of infection (during 32–34 weeks of infection) did not affect the mortality of the mice until 38 weeks postinfection (Fig 5J). Lung gross anatomy

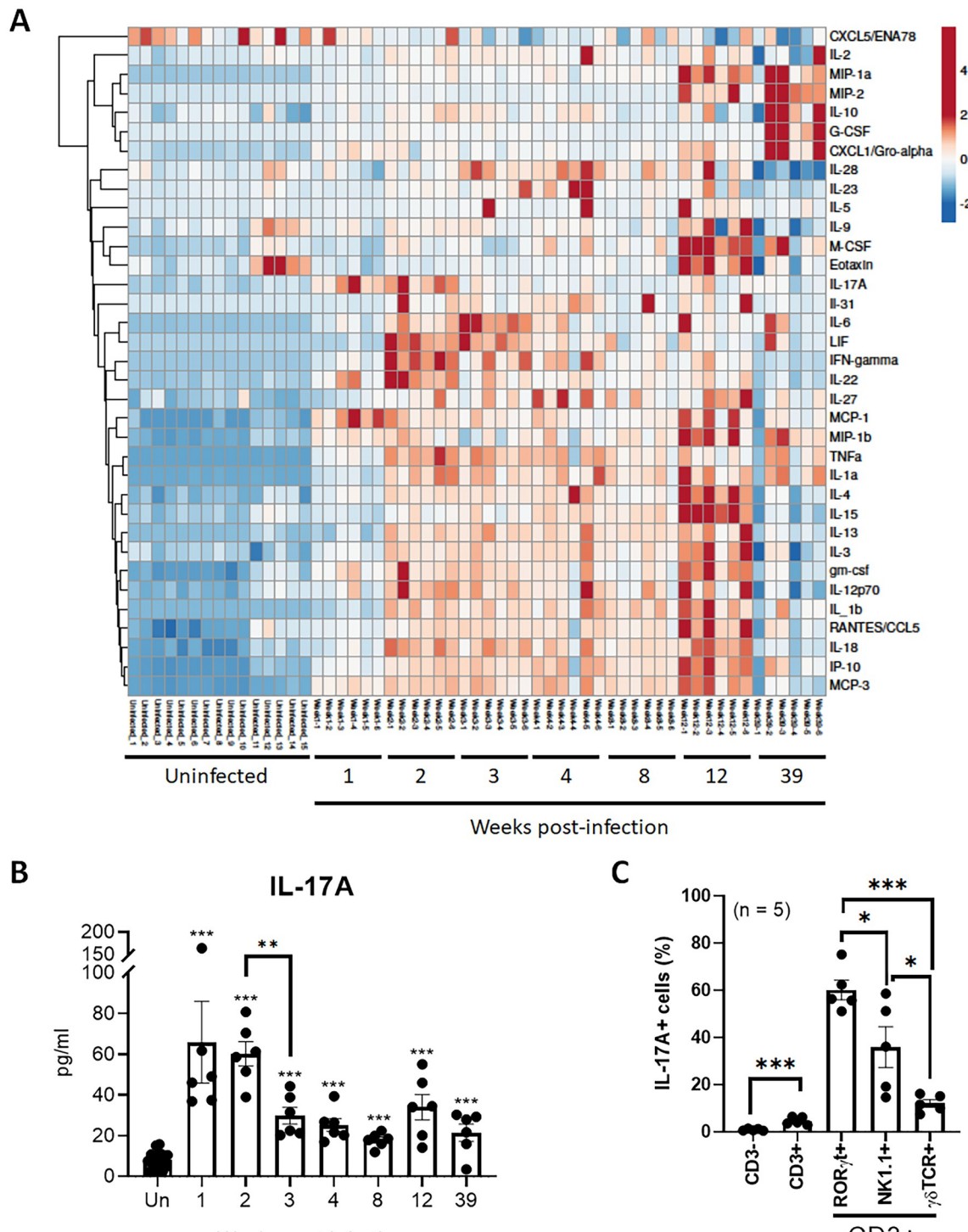

**Fig 4. Cytokine and chemokine profiles of *M. intracellulare*-infected mouse lungs. (A-B)** Lungs from uninfected control and *M. intracellulare*-infected mice were collected at 1, 2, 3, 4, 8, 12 and 39 weeks postinfection. Overall, 36 cytokines and chemokines were determined in lung homogenates of mice using multiplex ELISA. Data were pooled from two independent experiments (uninfected mice n = 15, infected mice n = 6 mice per indicated time point), and hierarchical clustering was performed (A). The IL-17A level at each time point is shown (B). **(C)** To identify the cellular source of IL-17A, lung cells were isolated from *M. intracellulare*-infected mice (n = 5) after 2 weeks of infection (because lung IL-17A levels peaked at 1–2 weeks postinfection), and IL-17A-producing cells were determined by flow cytometry. The percentages (%) of IL-17A-producing cells in CD45+CD3- cells, CD45+CD3+ cells, CD45+CD3+RORγt+ cells, CD45+CD3+NK1.1+ cells and CD45+CD3+γδTCR+ cells are shown (C). Data are expressed as the means ± SEM.

*P < 0.05, **P < 0.01, and ***P < 0.001. ns, not significant. P values for comparison between uninfected mice (Un) and other were indicated with asterisks sans connecting lines. For other comparisons, connecting lines used.

(Fig 5K) and histological images (S7B and S7C Fig) showed abnormal lung morphological changes in *M. intracellulare*-infected mice treated with either anti-Ly-6G mAb or isotype-matched control Ab compared with uninfected control mice at 38 weeks postinfection. However, no differences in the lung gross or histological lesions of anti-Ly-6G mAb- and isotype-matched control Ab-treated mice were noted (Fig 5K and S7D Fig). No differences in the bacterial burden in the lungs of anti-Ly-6G mAb- or isotype-matched control Ab-treated mice were found (Fig 5L). These findings suggest that late neutrophil infiltration of the lungs does not exert significant effects on mortality, lung damage or bacterial growth in chronically infected mice.

## IL-17A production during early *M. intracellulare* infection increases mortality, bacterial growth, and lung lesions in chronically infected mice

IL-17 is known to play an important role in protective immune responses against various bacterial infections [18]. However, excess IL-17 can cause immunopathology and death of the infected host [1,18]. We next tested our inference that neutralizing IL-17A favorably affects the survival, bacterial growth, and lung damage of mice infected with *M. intracellulare*. Fig 6A shows a schematic representation of the interventional approach using anti-IL-17A mAb or isotype-matched control Ab treatment of *M. intracellulare*-infected mice. One day after infection, the mice were treated with a neutralizing anti-IL-17A mAb or an isotype-matched control Ab every other day for 11 days. IL-17A levels were significantly reduced in the lungs of anti-IL-17A mAb-treated mice compared to isotype-matched control Ab-treated mice at 2 and 4 weeks postinfection (S9A and S9B Fig). As shown in Fig 6B, 55.34% (*P* < 0.05) of *M. intracellulare*-infected mice that received the isotype-matched control Ab died by 39 weeks postinfection. In contrast, 13.33% of mice that received the anti-IL-17A mAb died. As shown in S8B Fig, at 2 and 4 weeks postinfection, there was no significant difference in the bacterial burden of anti-IL-17A mAb- and isotype control Ab-treated mice. At 39 weeks postinfection, bacteria were significantly reduced noted in the lungs (Fig 6C. 7.71 ± 0.06 log vs. 6.83 ± 0.18 log CFU; *P* < 0.01), spleen (7.06 ± 0.20 log vs. 6.01 ± 0.22 log CFU; *P* < 0.05), MLN (5.50 ± 0.20 log vs. 4.70 ± 0.16 log CFU; *P* < 0.05), and liver (6.30 ± 0.28 log vs. 5.18 ± 0.3 log CFU; *P* < 0.05) of anti-IL-17A mAb-treated mice compared to isotype-matched control Ab-treated mice. Histological examination of lung tissue identified severe granulomatous (left panel of Fig 6D) and fibrotic lesions (left panel of Fig 6E) in isotype-matched control Ab-treated mice infected with *M. intracellulare*. However, early anti-IL-17A mAb treatment significantly reduced subsequent granulomatous (right panel of Fig 6D) and fibrotic changes (right panel of Fig 6E) in the lungs of *M. intracellulare*-infected mice. Early anti-IL-17A mAb treatment also increased the number of DCs (Fig 6G) and NK cells (Fig 6H) compared to isotype-matched Ab treatment in the lungs of *M. intracellulare*-infected mice.

## Anti-IL-17A antibody treatment reduces MMP-3 production in *M. intracellulare*-infected mice

Next, we determined the cytokine and chemokine levels of the infected mice treated with antibodies at 2, 4, and 39 weeks postinfection. Anti-IL-17A mAb or isotype-matched control Ab treatment was stopped at 11 days postinfection. Cytokine and chemokine levels, except IL-17A, were similar in both groups of mice at 2 and 4 weeks postinfection (S9 Fig). However, at

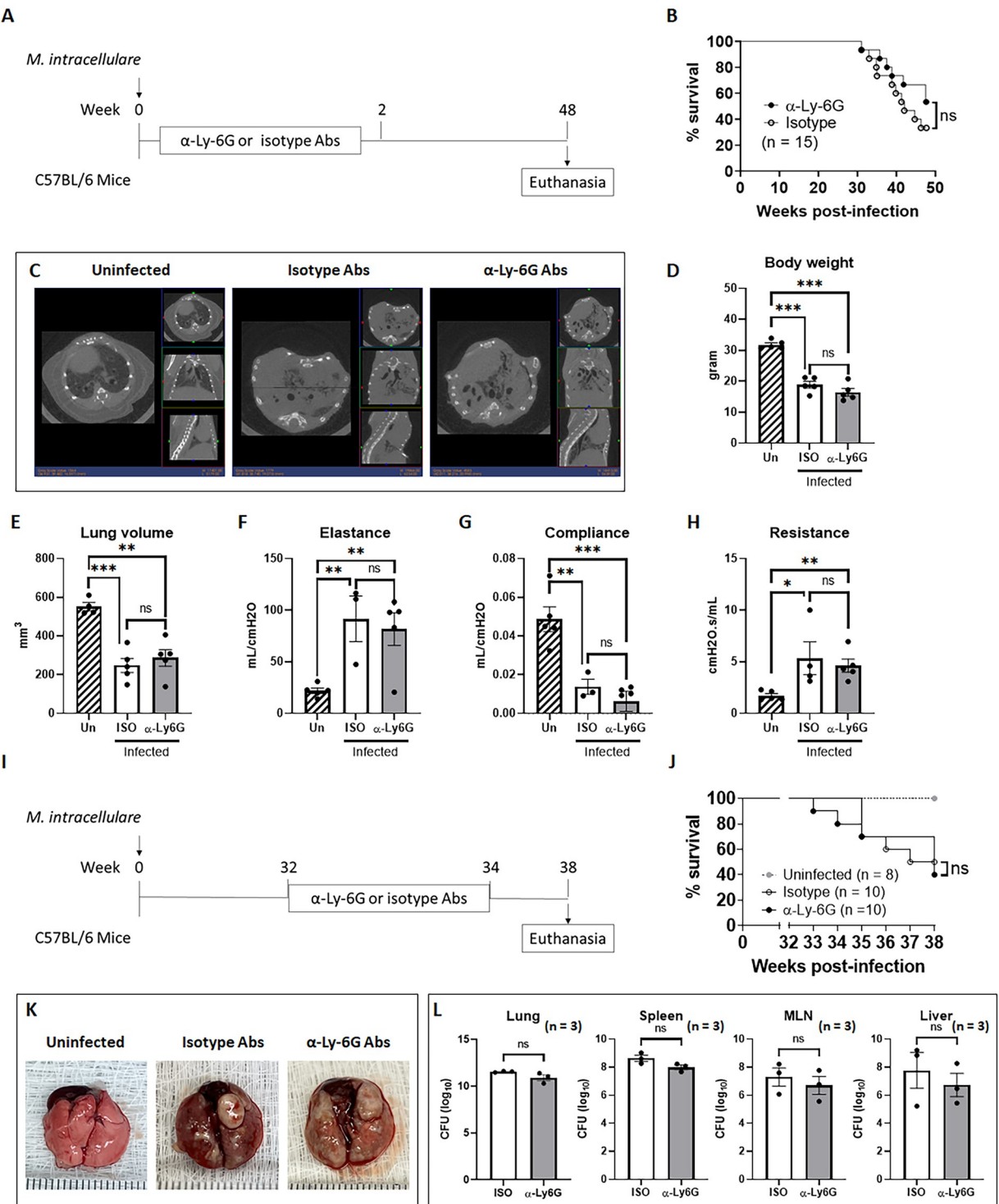

**Fig 5. Anti-Ly6G mAb treatment during early and chronic infection has no effect on mortality or lung damage in chronically infected mice.** (**A**) Schematic representation of *M. intracellulare* infection and the experimental schedule for early neutrophil depletion are shown. C57BL/6 mice were infected with clinical isolates of *M. intracellulare* ($5 \times 10^7$ CFU) via the intranasal route. To deplete neutrophils, *M. intracellulare*-infected mice were intraperitoneally injected with anti-Ly6G mAb or rat IgG2a isotype control Ab every other day from 1 to 11 days postinfection. (**B**) Survival curves for *M. intracellulare*-infected mice treated with anti-Ly6G mAb (black circle) or isotype control Ab (white circle) (B). Data were pooled from two independent experiments (total n = 15 mice per group). Survival curves were compared using the log rank test. ns, not significant. (**C-H**) Lung CT scanning and pulmonary function tests were conducted at 48 weeks postinfection. A representative lung CT scan image of each group is shown (C). Body weight (D), lung volume (E), lung elastance (F), lung compliance (G) and lung resistance (H) of

untreated and uninfected mice (Un, n = 4–5), isotype control Ab-treated (ISO, n = 3–5) or anti-Ly6G mAb-treated (α-Ly6G, n = 4–5) *M. intracellulare*-infected mice were evaluated at 48 weeks postinfection. (**I**) Schematic representation of *M. intracellulare* infection and the experimental design for late neutrophil depletion are shown. C57BL/6 mice were infected with clinical isolates of *M. intracellulare* at $5 \times 10^7$ CFU via the intranasal route. To deplete neutrophils, *M. intracellulare*-infected mice were intraperitoneally injected with anti-Ly6G mAb or rat IgG2a isotype control Ab 3 times per week from 32–34 weeks postinfection. (**J**) Survival curves for *M. intracellulare*-infected mice treated with anti-Ly6G mAb (black circle) or isotype control Ab (white circle) (J). Data were pooled from two independent experiments. Survival curves were compared using the log rank test. ns, not significant. (**K**) Representative lung gross morphology of uninfected control (left), *M. intracellulare*-infected and isotype control Ab-treated mice (middle) and *M. intracellulare*-infected and anti-Ly6G mAb-treated mice (right) at 38 weeks postinfection is shown. (**L**) Bacterial burden was determined in the lungs, spleen, mediastinal lymph nodes (MLNs), and liver at 38 weeks postinfection. Data are expressed as the means ± SEM. $^*P < 0.05$, $^{**}P < 0.01$, and $^{***}P < 0.001$. ns, not significant. Data from 2 independent experiments were combined.

39 weeks post-infection, anti-IL-17A mAb treatment significantly enhanced lung IL-4 (Fig 6I) and IL-22 (Fig 6J) levels compared to isotype-matched control Ab treatment in the lungs of *M. intracellulare*-infected mice. Early anti-IL-17A mAb treatment also significantly reduced the mRNA (Fig 6K) and protein levels (Fig 6M) of MMP-3, but MMP-9 in the lungs was decreased transiently and modestly early following *M. intracellulare* infection (Fig 6L, S11E and S11F Fig) compared with isotype Ab treatment.

## IL-17A exaggerates MMP-3 production by lung epithelial cells

Lung epithelial cells can be infected with *M. intracellulare* and are one of the major targets for NTM disease [15,31]. To determine the effect of IL-17A on lung epithelial cell elaboration of metalloproteinases, the lung epithelial cell line H1975 was infected at multiplicities of infection (MOIs) of 0, 100 and 1000 in the presence of IL-17A at 0, 100, 500 and 1000 ng/ml. MMP-3 (Fig 7A), MMP-9 (Fig 7B), bacterial burden (Fig 7C) and lactate dehydrogenase (LDH) activity (Fig 7D) were evaluated at 24 h, 48 h and 72 h postinfection. MMP-3 levels were significantly increased in the presence of IL-17A (above 100 ng/ml) upon infection with *M intracellulare* at 1000 MOI at 48 h and 72 h (Fig 7A). IL-17A did not affect the production of MMP-9 (Fig 7B), bacterial growth (Fig 7C) or cell viability (Fig 7D). These data suggest that IL-17A enhances production of MMP-3 by lung epithelial cells infected with *M. intracellulare* and does not affect the production of MMP-9, bacterial growth, or cell viability. To determine whether *M. intracellulare* infection affects the expression of IL-17 receptors, murine lungs were stained with an anti-IL-17RA antibody. As shown in S12A Fig, no difference in IL-17 receptor expression was noted between the control and *M. intracellulare*-infected mouse lung epithelium. IL-17RA expression was also comparable between *M. intracellulare*-infected H1975 human lung epithelial cells (1 day after *in vitro* infection) and uninfected cells (S12B Fig).

## Discussion

NTM can cause persistent infection and progressive lung damage in patients [1,9]. Therefore, establishment of a mouse model with persistent infection and progressive lung injury is required to better understand the pathogenesis of NTM. In the present study, we demonstrate that intranasal infection of clinical isolates of *M. intracellulare* causes persistent bacterial growth in the lung and extrapulmonary organs and progressive granulomatous and fibrotic lung damage in mice during the 39-week course of infection. This implies that our model is suitable to investigate the pathogenesis of NTM. We also found that *M. intracellulare* infection causes early production of IL-17A (during the 1st two weeks of infection) by T cells, especially RORγt+ T cells, which culminates in lung damage (inflammation and fibrosis) and death in chronically infected mice.

   Previous studies have sought to develop a chronic NTM mouse model. A low dose (less than $10^4$ CFU) of aerosol or intratracheal NTM administration did not cause persistent

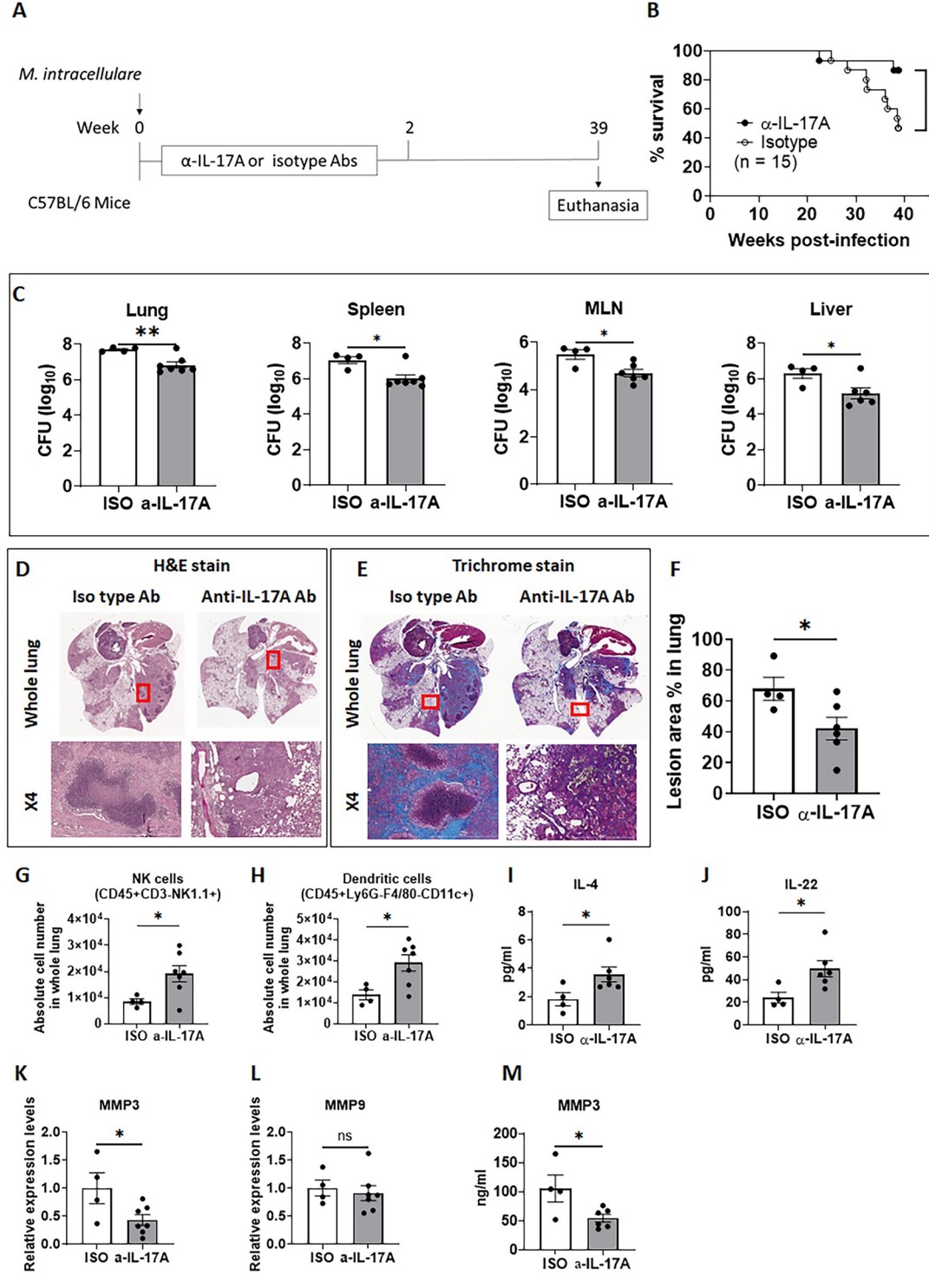

**Fig 6. Neutralization of IL-17A during early *M. intracellulare* infection reduces mortality, bacterial burden, and lung damage in chronically infected mice.** (**A**) Schematic representation of *M. intracellulare* infection and anti-IL-17A mAb or isotype control treatments is shown. Mice were intranasally administered normal saline or *M. intracellulare* suspension ($5 \times 10^7$ CFU in normal saline). To neutralize IL-17A, *M. intracellulare*-infected mice were intraperitoneally injected with 100 μg of anti-IL-17A mAb or 100 μg of mouse IgG1 isotype control Ab every other day from 1 to 11 days postinfection. (**B**) Survival curves for anti-IL-17A mAb (black circle)- or isotype control (white circle)-treated *M. intracellulare*-infected mice. Data were pooled from two independent experiments (total n = 15 mice per group). Survival curves were compared using the log rank test. (**C**) Bacterial burden was determined in the lungs, spleen, mediastinal lymph nodes (MLNs), and liver at 39 weeks postinfection. Data were

pooled from two independent experiments (total n = 4 mice in the isotype control Ab-treated group, total n = 7 mice in the anti-IL-17A mAb-treated group). (**D-F**) Lungs of the mice were formalin-fixed at 39 weeks postinfection (total n = 4 mice in isotype control Ab-treated group, total n = 6 mice in anti-IL-17A mAb-treated group). Paraffin-embedded tissue sections were prepared and stained with hematoxylin and eosin and trichrome. Representative images of H&E staining (D) and trichrome staining (E) of each group is shown. The lesion area (%) in lung images (F) was determined based on whole lung images stained with H&E at 39 weeks postinfection. (**G-M**) The immune cell subpopulation, cytokines/chemokines and MMPs were determined in lung homogenates of mice at 39 weeks postinfection (total n = 4 mice in the isotype control Ab-treated group, total n = 6 or 7 mice in the anti-IL-17A mAb-treated group). Dendritic cells (G) and NK cells (H) in the whole lung. IL-4 (I) and IL-22 (J) levels in lung homogenates. Relative mRNA expression levels of MMP-3 (K) and MMP-9 (L) and MMP-3 protein levels (M) in lung homogenates. Data are expressed as the means ± SEM. *P < 0.05, and **P < 0.01. ns, not significant. Data from 2 independent experiments were combined.

infection or progressive disease in mice [32,33]. A medium dose ($1 \times 10^5$ CFU) of *M. avium* strain 101 aerosol infection caused disseminated disease and increased bacterial burden at 8–12 weeks postinfection in BALB/c, C57BL/6, beige and nude mice, but bacterial burden was reduced in the lungs and spleens of all four strains of mice after 12 weeks of infection. None of the mice in this study appeared sick, and none died during the entire 23-week course of infection [34]. Intraperitoneal administration of a high dose ($5 \times 10^7$ CFU) of *M. avium* strain 104 caused systemic infection in C57BL/6 mice, and bacteria were not fully cleared by 25 weeks postinfection [35]. However, the bacterial burden was reduced in the lung and spleen at 12–17 weeks postinfection. Moreover, mice appeared healthy and showed no clinical signs of disease, such as fever, weight loss, loss of appetite, ruffled fur or discomfort. These findings suggested that suitable amounts of bacteria, clinical infection routes and bacterial strains are required to cause chronic NTM disease accompanied by persistent bacterial growth and a presentation similar to clinical disease in mice. In the current study, C57BL/6 mice were infected with a high dose ($5 \times 10^7$ CFU) of a clinical isolate of *M. intracellulare* via the intranasal route, and

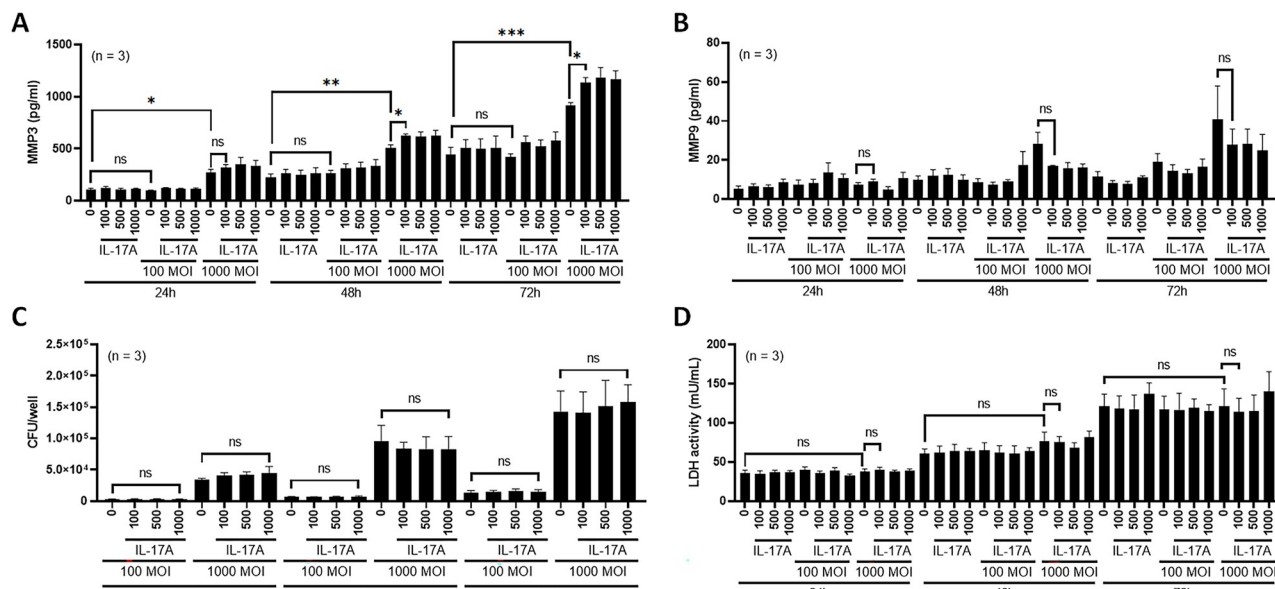

**Fig 7. Effects of IL-17A on *M. intracellulare*-infected human lung epithelial cells.** (**A-D**) The human lung epithelial cell line H1975 was infected with different bacterial numbers [0, 100 and 1000 multiplicity of infection (MOI)] in the presence of different concentrations of IL-17A (0, 100, 500 and 1000 ng/ml). MMP-3 (A), MMP-9 (B), bacterial burden (C) and lactate dehydrogenase (LDH) activity (D) were evaluated at the indicated time points. MMP-3 (A), MMP-9 (B) and LDH activity (D) were evaluated in cell culture supernatants, and bacterial burden (C) was determined in cell lysates. Data are expressed as the means ± SEM. *P < 0.05, **P < 0.01, and ***P < 0.001. ns, not significant. Three independent experiments were performed.

weight loss, persistent bacterial growth, and progressive lung damage were observed in the infected mice during the 39-week infection. These findings indicate that we successfully established a chronic *M. intracellulare* mouse model. However, there are limitations of this mouse model. In humans, there are two major categories of NTM diseases. Most cases of NTM infection occur with isolated lung disease and extrapulmonary-disseminated disease is relatively rare [9]. However, disseminated infection is a common phenomenon in NTM mouse models [36], indicating that it is difficult to mimic chronic human NTM disease that is exclusively restricted to the lungs. Our mouse model has characteristics of both pulmonary and disseminated infection. This may be because of discrepancies between humans and mice. In the current study, we mainly focused on host lung immune responses from acute to chronic phases of *M. intracellulare* infection. Future studies should be conducted on the disease stage of interest.

Excessive immune cell infiltration at the site of infection leads to proinflammatory cytokine and chemokine production and lung damage [37]. Significant immune cell infiltration was noted during the first 2 weeks of infection (Fig 3). Among various cytokines and chemokines, IL-17A rapidly increased during the first week of infection (Fig 4A and 4B). Although NK cells and neutrophils were rapidly increased in the lungs during the first 2 weeks of infection, CD3 + cells, especially RORγt+ cells, represented the major source of IL-17A (Fig 4C and S3 Fig). In mice, NK cells are essential for protection against *M. kansasii* [38]. Depletion of NK cells increased mortality and lung bacterial burden in C57BL/6 mice infected with *M. kansasii* [38]. The number of NK cells in the lungs of infected mice was reduced after 3 weeks of infection, and bacterial burden was increased, suggesting that NK cells failed to control early infection. Neutrophils have been described as critical immune cells in the front line of defense against bacterial infection [39]. However, neutrophil depletion had no effect on mortality or bacterial growth in mice infected with *M. tuberculosis* Erdman, *M. tuberculosis* CDC1551, *M. bovis* BCG and *M. fortuitum*, whereas it increased mortality and bacterial growth in mice infected with *Salmonella* typhimurium and *Listeria monocytogenes* [40], suggesting that the role of neutrophils may vary by pathogen. In the present study, lung neutrophils were significantly increased at 1 week postinfection, gradually reduced after 2 weeks postinfection and increased again at 39 weeks postinfection. This coincided with mortality, suggesting that neutrophil infiltration may contribute to the mortality of infected mice. To determine the role of early and late neutrophils in mortality, *M. intracellulare*-infected mice were treated with anti-Ly6G mAb or isotype-matched control Ab during the first 2 weeks (0–2 weeks postinfection; Fig 5A–5H) or last 2 weeks (32–34 weeks of infection; Fig 5I–5L) of infection. Unexpectedly, treatment with an anti-Ly6G mAb during early or late infection had no effect on mortality, lung damage or bacterial growth in chronically infected mice despite clear depletion of neutrophils. While we did not deplete neutrophils during the entire course of infection, these data suggest that neutrophils during early and late infection do not contribute substantively to mortality or lung damage of mice chronically infected with *M. intracellulare*.

We found that IL-17A levels rapidly increased in the lungs of infected mice at 1–2 weeks postinfection (Fig 4A and 4B). Although IL-17 protects against various respiratory pathogens, it is also implicated in the induction of inflammatory lung damage [18]. Mice with defective IL-17 or IL-17RA expression are more susceptible to various respiratory pathogens, including *Streptococcus pneumoniae* [41], *Pseudomonas aeruginosa* [42] and *M. tuberculosis* HN878 [43], whereas IL-17-producing T cells are associated with immunopathology and inflammation in patients with multidrug-resistant TB [44]. Moreover, IL-17A and IL-17A-mediated immune responses are highly related to lung damage caused by aberrant inflammation in various pulmonary diseases, such as COPD [13] and CF [14]. In *M. intracellulare*-infected mice, IL-17A levels were significantly elevated during the first 2 weeks of infection (Fig 4A and 4B). Anti-IL-17A mAb treatment during this period of infection reduced immune cell infiltration at early

stages of infection (2 and 4 weeks postinfection) (S8C Fig) and reduced lung fibrosis, bacterial burden and mortality at chronic stages of infection (after 39 weeks of infection). The findings demonstrate that early production of IL-17A helps establish *M. intracellulare* infection.

IL-17A is critically involved in lung injury and fibrosis [18]. IL-17 increases MMP-3 activity in human epithelial cells [45], and MMP-3 is one of the major mediators of pulmonary fibrosis [29]. IL-17RA is expressed in the lung epithelium of both mice and humans (S12 Fig). MMP-3, but not MMP-9, production was reduced by anti-IL-17A mAb treatment in *M. intracellulare*-infected mice at 39 weeks postinfection (Fig 6K–6M) and MMP-3 production by *M. intracellulare*-infected lung epithelial cells (H1975 cells) was increased in the presence of IL-17A (Fig 7A). Hence, our findings suggest that IL-17A enhances pulmonary inflammation and fibrosis in *M. intracellulare*-infected mice, at least in part through MMP-3 production.

Early anti-IL-17A mAb treatment significantly increased the number of NK cells and DCs in the lungs of *M. intracellulare*-infected mice during chronic infection (39 weeks postinfection). DC-NK crosstalk plays an important role in the induction of optimal protective immune responses against microbial infections [46], including *M. tuberculosis* infection, through macrophage activation [47]. Anti-IL-17A mAb treatment also enhanced the IL-4 and IL-22 levels in the lungs of *M. intracellulare*-infected mice at 39 weeks postinfection. IL-4, an anti-inflammatory cytokine that mainly suppresses the proinflammatory milieu, enhances *M. tuberculosis* growth in mouse lungs [48,49]. However, it is unclear whether increased IL-4 plays any role in reducing inflammation in *M. intracellulare*-infected mice. IL-22 produced by human NK cells inhibits the growth of *M. tuberculosis* in human monocyte-derived macrophages [50]. The absence of IL-22 results in increased susceptibility to *M. tuberculosis* infection during the chronic stages of infection, but IL-22 is dispensable during the acute stages [51]. IL-22 also inhibits pulmonary fibrosis [52]. In the current study, we found that early IL-17A production reduces DCs, NK cell and IL-22 production in the lungs of mice chronically infected with *M. intracellulare*. Further investigation may provide additional important information about IL-17A-mediated regulation of NK-DC interactions and early establishment of pulmonary *M. intracellulare* infection. This work suggests that strategies targeting IL-17 could mitigate pulmonary damage associated with *M. intracellulare* infection.

Collectively, IL-17A production during early *M. intracellulare* infection reduced lung DCs and NK cell numbers and IL-22 production and was associated with increased mortality, bacterial growth, lung lesions and MMP-3 production in chronically infected mice. Moreover, exogenous IL-17A exaggerated MMP-3 production by lung epithelial cells upon *M. intracellulare* infection. These findings suggest that early IL-17A production precedes and promotes organizing pulmonary *M. intracellulare* infection in mice, at least in part through MMP-3 production. Our findings provide important new insights regarding the role of IL-17A in the pathogenesis of *M. intracellulare* infection and suggest a new candidate interventional strategy targeting IL-17A to reduce pulmonary injury in *M. intracellulare*-infected patients.

## Materials and methods

### Animals and ethics statement

Specific pathogen-free female C57BL/6 mice (6 to 8 weeks old) were purchased from Jackson Laboratory and housed at the animal facility at the University of Texas Health Science Center at Tyler. All animal studies were approved by the Institutional Animal Care and Use Committee (IACUC) of the University of Texas Health Science Center at Tyler (protocol #650). All animal procedures involving the care and use of mice were performed in accordance with the guidelines of the National Institutes of Health (NIH)/Office of Laboratory Animal Welfare (OLAW).

## Bacterial source and stock

*M. intracellulare* was isolated from a patient with NTM (patient #16:AF399) at the University of Texas Health Science Center at Tyler. This bacterial isolate is resistant to amikacin and clarithromycin. *M. intracellulare* was grown to mid-log phase (optical density was 0.4–0.6 at 600 nm) in 7H9 broth medium supplemented with 0.2% glycerol and 10% ADC enrichment (complete medium for bacterial culture) and harvested by centrifugation at 4,000 rpm for 25 min. The bacterial pellet was resuspended and adjusted to an optical density of 1 at 600 nm with fresh complete medium for bacterial culture. The bacterial suspension was mixed with an equal volume of 50% glycerol, aliquoted, and frozen at -80˚C until use. The concentration of aliquoted bacteria was confirmed by plating serially diluted bacterial suspensions onto 7H10 agar supplemented with 0.2% glycerol and 10% OADC enrichment and counting CFUs after 2 weeks of incubation, yielding $8.33 \pm 0.07 \log_{10}$ ($2.15 \times 10^8 \pm 0.35$) CFU/ml.

## Intranasal infection of mice with *M. intracellulare*

For infection, the bacterial stock was washed and resuspended in sterile normal saline (1 ml stock bacteria were resuspended in a final volume of 200 μl of sterile normal saline). A total of $7.73 \pm 0.08 \log_{10}$ ($5.4 \times 10^7 \pm 0.9$) CFU in a 50-μl volume of the bacterial suspension was intranasally administered to mice that were completely anesthetized with a mixture of ketamine and xylazine. Initial inoculated live bacteria counts were determined one day postinfection by plating serial dilutions of lung homogenates of mice onto 7H10 agar supplemented with 0.2% glycerol and 10% OADC enrichment. Two weeks after incubation, the CFU was $7.51 \pm 0.08 \log_{10}$ ($3.3 \times 10^7 \pm 0.6$)/lung.

## Determination of bacterial growth in mouse organs

To determine bacterial growth, the mice were sacrificed at the indicated time points after infection. The lungs, MLNs, spleens and livers were removed aseptically and homogenized in PBS (lungs, spleens, and livers in 2 ml of PBS and MLNs in 1 ml of PBS). The bacterial burden in each organ was determined by plating serial dilutions of each organ homogenate as described above.

## Determination of mortality in mice infected with *M. intracellulare*

Mice were observed daily for signs of distress (ruffled fur, weight loss, reluctance to move, ataxia and labored respiration). For mortality studies, mice that lost 20% of their body weight were assessed by the attending veterinarian or vivarium staff to determine whether mice needed to be euthanized. Mortality was recorded until 50%–75% of mice had died.

## H&E and trichrome staining and histopathological analysis

For histopathological analysis, mice were sacrificed at the indicated time points, and whole lungs fixed in 10% neutral-buffered formalin, embedded in paraffin, and sectioned at a thickness of 5 μm. The tissue sections were deparaffinized and rehydrated with xylene and a series of graded alcohol and then stained with hematoxylin and eosin (H&E) or Masson's trichrome. The severity of lung inflammation was quantified as previously described [53] using a score from 0 (no inflammation) to 4 (severe inflammation) for each of the following criteria: alveolar wall inflammation, alveoli destruction, leukocyte infiltration, and perivascular inflammation. The severity of lung fibrosis was quantified according to the Ashcroft scoring system [54]. All histopathological evaluations were performed in a blinded fashion. Data from two independent experiments were combined.

### Immunohistochemical staining

Immunohistochemical staining was performed with mouse lung sections using a commercial detection system (TP-015-HA, Thermo Scientific) as previously described [55]. Heat-induced epitope retrieval was performed by placing the slides in a glass slide container with citrate buffer (0.1 M citric acid and 0.1 M sodium citrate, pH 6.0) in a water bath at 95°C for 20 min. After cooling to room temperature, the sections were blocked by serial incubation with hydrogen peroxide, Ultra V block, and 5% goat serum in PBS. The blocked lung sections were incubated with anti-IL-17RA Ab (PA5-34571, Invitrogen) or rabbit IgG isotype control (08–6199, Invitrogen) overnight at 4°C in a humidified chamber. After incubation with biotinylated goat anti-mouse IgG and streptavidin-HRP, the sections were rinsed with PBS and incubated with 3-amino-9-ethylcarbazole, a chromogenic HRP substrate, followed by counterstaining with hematoxylin.

### Immune cell phenotyping by flow cytometry

To identify lung immune cell subpopulations, the mice were sacrificed at the indicated time points, and the whole lung of each mouse was mechanically homogenized and filtered through a 40-μm cell strainer. Red blood cells (RBCs) were lysed using RBC lysing buffer (Sigma–Aldrich) for 3 min on ice, and surface staining for leukocyte subpopulations was performed with different combinations of antibodies (listed in S1 Table) for 30 min on ice. The cells were washed and resuspended in PBS, and acquisition was performed using an Attune NxT acoustic flow cytometer (Invitrogen). FlowJo software version 10 (BD) was used for flow cytometry data analysis. The gating strategy for mouse lung leukocyte subpopulations is shown in S1 Fig.

### Determination of the cellular source of IL-17A

To identify the cellular source of IL-17A, lung cells of *M. intracellulare*-infected mice were prepared 2 weeks postinfection (because lung IL-17A levels peaked at 1–2 weeks postinfection) as described above. For the intracellular accumulation of IL-17A, lung cells were incubated with PMA (1 μg/ml), ionomycin (1 μg/ml) and brefeldin A (5 μg/ml) for 4 h at 37°C as previously described [56]. Surface staining was performed with anti-CD45 antibody (103116, Biolegend), CD3 (100312, Biolegend), NK1.1 (108745, Biolegend) and γδTCR (118124, Biolegend) as described above. Intracellular staining for IL-17A and RORγt (562894, BD) was performed using a commercial transcription factor buffer set (424401, Biolegend) according to the manufacturer's instructions. Data acquisition and analysis of flow cytometry data were performed as described above. The gating strategy for identifying IL-17A-producing cells is shown in S3 Fig.

### Multiplex ELISA to measure cytokine and chemokine levels in lung homogenates

Mouse multiplex ELISA kits (36-Plex kits, Invitrogen) were used to measure the chemokine and cytokine levels in lung homogenates according to the manufacturer's instructions.

### Hierarchical clustering

Expression values of cytokines/chemokines from uninfected and infected mice were mean-centered for each cytokine/chemokine. Hierarchical clustering was performed using the R programming environment (R version 4.1.0).

### Depletion of neutrophils and IL-17A neutralization

To deplete immune cells and neutralize cytokines, mice were injected intraperitoneally with each antibody as previously described [57] with some modifications. To deplete neutrophils

during early infection, mice infected with *M. intracellulare* were intraperitoneally injected with an initial dose of 400 µg followed by 300 µg of anti-Ly6G antibody (clone 1A8, BioXcell) or an initial dose of 400 µg followed by 300 µg of rat IgG2a isotype control (clone 2A3, BioX-cell) every other day from 1 to 11 days postinfection (Fig 5A). To deplete neutrophils during chronic infection, mice infected with *M. intracellulare* were intraperitoneally injected with an initial dose of 400 µg followed by 300 µg of anti-Ly6G antibody (clone 1A8, BioXcell) or an initial dose of 400 µg followed by 300 µg of rat IgG2a isotype control (clone 2A3, BioXcell) 3 times per week for 2 weeks from 32 to 34 weeks postinfection (Fig 5I). To neutralize IL-17A, mice infected with *M. intracellulare* were intraperitoneally injected with 100 µg of anti-IL-17A antibody (clone 17F3, BioXcell) or 100 µg of mouse IgG1 isotype control (clone MOPC-21, BioXcell) every other day from 1 to 11 days postinfection.

## Pulmonary function testing

Pulmonary function tests were performed immediately before computed tomographic (CT) imaging and before sacrifice, as previously described [58–61]. Briefly, mice were anesthetized with a ketamine/xylazine mixture, and anesthetized mice were intubated by inserting a sterile, 20-gauge intravenous cannula through the vocal cords into the trachea. Mice were maintained under anesthesia using isoflurane during pulmonary function testing. Measurements were then performed using the flexiVent system (SCIREQ, Tempe, AZ). The snapshot perturbation method was used to determine lung compliance according to the manufacturer's specifications.

## CT scans and measurements of lung volume

Chest CT imaging and measurements of lung volumes were performed as previously described [58–61]. Mice anesthetized with ketamine/xylazine were maintained under anesthesia using an isoflurane/oxygen mixture to minimize spontaneous breaths throughout the procedure. Images were obtained using the Explore Locus Micro-CT Scanner (GE Healthcare, Waukesha, WI). CT scans were performed at a resolution of 93 mm. Microview software version 2.2 (http://microview.sourceforge.net) was used to analyze lung volumes and to render three-dimensional images. Lung volumes were calculated from renditions collected at full inspiration.

## Evaluation of relative MMP-3 and MMP-9 mRNA expression levels in lung homogenates

Total RNA was extracted from lung homogenates using TRIzol LS reagent (Invitrogen). MMP-3 and MMP-9 mRNA expression levels were measured by real-time PCR from the total RNA using GAPDH as an internal control with specific primer and probe sets (Applied Biosystems).

## Evaluation of MMP-3 protein levels in lung homogenates

Mouse MMP-3 ELISA kits (R&D Systems) were used to measure the MMP-3 levels in lung homogenates according to the manufacturer's instructions.

## Determination of IL-17RA expression, bacterial burden, MMP-3, MMP-9 and LDH in lung epithelial cells

H1975 cells ($5 \times 10^4$ cells per well for 24-well plates, $2 \times 10^5$ cells per well for 6-well plates) were plated in antibiotic-free RPMI-1640 containing 5% heat-inactivated fetal bovine serum

(FBS). After one day of incubation, the cells were washed 2 times with fresh RPMI-1640 containing 5% heat-inactivated FBS and incubated with *M. intracellulare* at 0, 100 and 1000 multiplicity of infection (MOI) at 37˚C in 5% $CO_2$. After a 2-h incubation, the cells were washed three times with fresh RPMI-1640 containing 5% heat-inactivated FBS to remove extracellular bacilli as previously described [62]. The initial infected live bacterial counts were determined by plating serial dilutions of cell lysates onto 7H10 agar supplemented with 0.2% glycerol and 10% OADC enrichment. H1975 cells were further incubated in the presence of IL-17A at 0, 100, 500 and 1000 ng/ml at 37˚C in 5% $CO_2$. After 1 day of infection, IL-17RA expression was determined from the lysate of H1975 cells by western blot using an anti-IL-17RA polyclonal antibody (Invitrogen). Bacterial burden and MMP-3, MMP-9 and LDH levels were evaluated at 24 h, 48 h and 72 h postinfection. Bacterial burden was determined from cell lysates. MMP-3, MMP-9 and LDH were evaluated in the cell culture supernatant of H1975 cells using commercial ELISA kits (R&D Systems) and colorimetric LDH assay kits (Abcam). Three independent experiments were performed, and data from these experiments were combined.

## Statistical analyses

The results are expressed as the means ± SEM. Comparisons between groups were performed using an unpaired t test (two-tailed). Mouse survival was compared using the Kaplan–Meier log-rank test. $P < 0.05$ was considered significant.

## Supporting information

**S1 Fig. Gating strategy for immune cell phenotyping.** Mouse lung cells were stained with various antibodies (S1 Table), and immune cells were determined using the illustrated gating strategy. Neutrophils (Ne) were gated as CD45+CD11b+Ly6G+ cells. Interstitial macrophages (IMs) were gated as CD45+Ly6G-F4/80+CD11c-cells. Alveolar macrophages (AMs) were gated as CD45+Ly6G-F4/80+CD11c+ cells. Dendritic cells (DCs) were gated as CD45+-Ly6G-F4/80-CD11c+ cells. B cells were gated as CD45+Ly6G-CD19+CD3- cells. T cells were gated as CD45+Ly6G-CD19-CD3+ cells. CD8 T (Tc) cells were gated as CD45+-Ly6G-CD19-CD3+CD8+CD4- cells. CD4 T (Th) cells were gated as CD45+Ly6G-CD19-CD3+CD8-CD4+ cells. NK cells were gated as CD45+CD3-NK1.1+ cells.
(TIF)

**S2 Fig. Cytokine and chemokine profiles in the lungs of *M. intracellulare*-infected mice.** C57BL/6 mice were infected with clinical isolates of *M. intracellulare* ($5 \times 10^7$ CFU) via the intranasal route. Mice were sacrificed at 1, 2, 3, 4, 8, 12 and 39 weeks postinfection. Lungs were collected from uninfected control and *M. intracellulare*-infected mice, and 36 cytokine and chemokine levels were determined in lung homogenates of mice using multiplex ELISAs. IFN-α was under the limit of detection. Data were pooled from two independent experiments (uninfected mice n = 15, infected mice n = 6 mice per indicated time point). Data are expressed as the means ± SEM. *P < 0.05, **P < 0.01, and ***P < 0.001. ns, not significant. P values for comparison between uninfected mice (Un) and other were indicated with asterisks sans connecting lines. For other comparisons, connecting lines used.
(TIF)

**S3 Fig. Gating strategy for IL-17A-producing cells.** Mouse lung cells were stained with various antibodies (S1 Table), and the percentages (%) of IL-17A-producing cells in CD45+CD3-cells, CD45+CD3+ cells, CD45+CD3+RORγt+ cells, CD45+CD3+NK1.1+ cells and CD45+CD3+γδTCR+ cells were determined using the illustrated gating strategy.
(TIF)

**S4 Fig. Body weight changes in mice infected with *M. intracellulare*.** (**A-D**) C57BL/6 mice were infected with clinical isolates of *M. intracellulare* ($5 \times 10^7$ CFU) via the intranasal route. Body weight was monitored in *M. intracellulare*-infected mice (A), *M. intracellulare*-infected mice treated with anti-Ly6G mAb or isotype-matched control Ab in early infection (B), *M. intracellulare*-infected mice treated with anti-Ly6G mAb or isotype-matched control Ab in chronic infection (C) and *M. intracellulare*-infected mice treated with anti-IL-17A mAb or isotype-matched control Ab (D). Data are expressed as the means ± SEM. $^*$P < 0.05, $^{**}$P < 0.01, and $^{***}$P < 0.001 compared with initial body weight (A) or between anti-IL-17A mAb- or isotype-matched control Ab-treated mice (D).
(TIF)

**S5 Fig. Bacterial burden, immune cell subpopulations, and histology in *M. intracellulare*-infected mice treated with anti-Ly6G mAb or isotype-matched control Ab during the first 4 weeks of infection.** (**A**) Schematic representation of *M. intracellulare* infection and the experimental schedule for early neutrophil depletion are shown. C57BL/6 mice were infected with clinical isolates of *M. intracellulare* ($5 \times 10^7$ CFU) via the intranasal route. To deplete neutrophils, *M. intracellulare*-infected mice were intraperitoneally injected with anti-Ly6G mAb or rat IgG2a isotype control Ab every other day from 1 to 11 days postinfection. Lungs were collected at 2 and 4 weeks postinfection. (**B**) Bacterial burden in the lung, spleen, mediastinal lymph node (MLN) and liver was determined. (**C**) The absolute number of immune cells per whole lung was determined by flow cytometry. (**D**) Representative figures of lung histology are shown. (**E**) The severity of lung inflammation was quantified from a total of 4 mice per indicated time point using a score from 0 (no inflammation) to 4 (severe inflammation) for each of the following criteria: alveolar wall inflammation, alveolar destruction, leukocyte infiltration, and perivascular inflammation. Data were pooled from two independent experiments. Data are expressed as the means ± SEM. $^*$P < 0.05, $^{**}$P < 0.01, and $^{***}$P < 0.001.
(TIF)

**S6 Fig. Cytokine and chemokine levels in the lungs of *M. intracellulare*-infected mice treated with anti-Ly6G mAb or isotype-matched control Ab during early infection.** Cytokine and chemokine levels were determined in lung homogenates of mice at 2 weeks postinfection using multiplex ELISA. IFN-α was under limit of detection. Data were pooled from two independent experiments. Data are expressed as the means ± SEM. $^*$P < 0.05, and $^{**}$P < 0.01.
(TIF)

**S7 Fig. Immune cell subpopulation and histology in *M. intracellulare*-infected mice treated with anti-Ly6G mAb or isotype-matched control Ab during late infection.** To deplete neutrophils during chronic infection, mice infected with *M. intracellulare* were intraperitoneally injected with an initial dose of 400 μg followed by 300 μg of anti-Ly6G antibody (clone 1A8, BioXcell) or an initial dose of 400 μg followed by 300 μg of rat IgG2a isotype control (clone 2A3, BioXcell) 3 times per week for 2 weeks from 32 to 34 weeks postinfection. (**A**) The absolute number of immune cells per whole lung was determined at 38 weeks postinfection by flow cytometry. (**B-D**) Representative images of H&E staining (B) and trichrome staining (C) of each group are shown. The lesion area (%) in lung images (D) was determined based on whole lung images stained with H&E at 38 weeks postinfection. Data were pooled from two independent experiments. Data are expressed as the means ± SEM. $^*$P < 0.05, $^{**}$P < 0.01, and $^{***}$P < 0.001. ns, not significant.
(TIF)

**S8 Fig. Bacterial burden, immune cell subpopulation, and histology in *M. intracellulare*-infected mice treated with anti-IL-17A mAb or isotype-matched control Ab during the**

**early 4 weeks of infection.** (**A**) Schematic representation of *M. intracellulare* infection and the experimental schedule for neutralization of IL-17A during early infection are shown. C57BL/6 mice were infected with clinical isolates of *M. intracellulare* ($5 \times 10^7$ CFU) via intranasal delivery. To neutralize IL-17A, *M. intracellulare*-infected mice were intraperitoneally injected with anti-IL-17A mAb or IgG1 isotype control Ab every other day from 1 to 11 days postinfection. Lungs were collected at 2 and 4 weeks postinfection. (**B**) Bacterial burden in the lung, spleen, mediastinal lymph node (MLN) and liver was determined. (**C**) The absolute number of immune cells per whole lung was determined by flow cytometry. (**D**) Representative figures of lung histology are shown. (**E**) The severity of lung inflammation was quantified in 4 mice at each indicated time point using a score from 0 (no inflammation) to 4 (severe inflammation) for each of the following criteria: alveolar wall inflammation, alveolar destruction, leukocyte infiltration, and perivascular inflammation. Data were pooled from two independent experiments. Data are expressed as the means ± SEM. $^*P < 0.05$. ns, not significant.
(TIF)

**S9 Fig. Cytokine and chemokine levels in the lungs of *M. intracellulare*-infected mice treated with anti-IL-17A mAb or isotype-matched control Ab at 2 and 4 weeks postinfection.** (**A, B**) Cytokine and chemokine levels were determined in lung homogenates of *M. intracellulare*-infected mice treated with anti-IL-17A mAb or isotype-matched control Ab at 2 weeks (A) and 4 weeks postinfection (B) using multiplex ELISA. IFN-α was under limit of detection. Data were pooled from two independent experiments. Data are expressed as the means ± SEM. $^*P < 0.05$ and $^{**}P < 0.01$.
(TIF)

**S10 Fig. Immune cell subpopulation and cytokine/chemokine levels in the lungs of *M. intracellulare*-infected mice treated with anti-IL-17A mAb or isotype-matched control Ab at 39 weeks postinfection.** (**A, B**) The absolute number of immune cells per whole lung was determined in lung homogenates of *M. intracellulare*-infected mice treated with anti-IL-17A mAb or isotype-matched control at 39 weeks postinfection by flow cytometry (A). Cytokine and chemokine levels were determined in lung homogenates of mice at 39 weeks postinfection using multiplex ELISA (B). IFN-α was under limit of detection. Data were pooled from two independent experiments. Data are expressed as the means ± SEM. $^*P < 0.05$.
(TIF)

**S11 Fig. Levels of MMP-3 and MMP-9 in the lungs of *M. intracellulare*-infected mice treated with anti-IL-17A mAb or isotype-matched control Ab at 2 and 4 weeks postinfection.** (**A-F**) RNA expression levels (A, B) and protein levels (C, D) of MMP-3 and RNA expression levels of MMP-9 (E, F) were determined from the lungs of *M. intracellulare*-infected mice treated with anti-IL-17A mAb or isotype-matched control at 2 and 4 weeks postinfection. Data were pooled from two independent experiments (total n = 6 mice per group at each indicated time point). Data are expressed as the means ± SEM. $^*P < 0.05$, $^{**}P < 0.01$, and $^{***}P < 0.001$.
(TIF)

**S12 Fig. Expression of IL-17RA on mouse lung and human lung epithelial cells.** (**A**) Immunohistochemical staining was performed to determine the expression of IL-17RA in the lungs of uninfected mice or *M. intracellulare*-infected mice after 2 weeks of infection. (**B**) IL-17RA expression was determined from the lysate of H1975 cells after 1 day of infection by western blot. (**C**) The intensity of each target protein band was normalized to that of β-actin. Three independent experiments were performed. Data are expressed as the means ± SEM.
(TIF)

**S1 Table. Antibodies for flow cytometry analysis.**
(XLSX)

## Author Contributions

**Conceptualization:** Bock-Gie Jung, Ramakrishna Vankayalapati.

**Data curation:** Bock-Gie Jung.

**Formal analysis:** Bock-Gie Jung, Ramakrishna Vankayalapati.

**Funding acquisition:** Richard J. Wallace, Jr., Barbara A. Brown-Elliott, Steven Idell, Julie V. Philley, Ramakrishna Vankayalapati.

**Investigation:** Bock-Gie Jung, Kristin Dean.

**Methodology:** Bock-Gie Jung, Ramakrishna Vankayalapati.

**Project administration:** Bock-Gie Jung, Ramakrishna Vankayalapati.

**Resources:** Bock-Gie Jung, Ramakrishna Vankayalapati.

**Software:** Bock-Gie Jung.

**Supervision:** Bock-Gie Jung, Ramakrishna Vankayalapati.

**Validation:** Bock-Gie Jung, Ramakrishna Vankayalapati.

**Visualization:** Bock-Gie Jung, Ramakrishna Vankayalapati.

**Writing – original draft:** Bock-Gie Jung, Ramakrishna Vankayalapati.

**Writing – review & editing:** Bock-Gie Jung, Buka Samten, Richard J. Wallace, Jr., Barbara A. Brown-Elliott, Torry Tucker, Steven Idell, Julie V. Philley, Ramakrishna Vankayalapati.

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
