## [Decision Letter · Decision Letter 0]

11 Oct 2021

Dear Dr. Jung,

Thank you very much for submitting your manuscript "Early IL-17A production helps establish Mycobacterium intracellulare infection in mice" for consideration at PLOS Pathogens. As with all papers reviewed by the journal, your manuscript was reviewed by members of the editorial board and by several independent reviewers. In light of the reviews (below this email), we would like to invite the resubmission of a significantly-revised version that takes into account the reviewers' comments. Overall, the reviewers concur that this is an important study but that it requires additional experiments and data analysis to fully justify the conclusions.  Specifically, i) in vitro data need to be strengthened to establish the link between IL-17 and MMP3; ii) blocking neutrophils at a later time point; and i) additional analysis of the already generated data. 

We cannot make any decision about publication until we have seen the revised manuscript and your response to the reviewers' comments. Your revised manuscript is also likely to be sent to reviewers for further evaluation.

Sincerely,

Padmini Salgame

Associate Editor

PLOS Pathogens

JoAnne Flynn

Section Editor

PLOS Pathogens

Kasturi Haldar

Editor-in-Chief

PLOS Pathogens

orcid.org/0000-0001-5065-158X

Michael Malim

Editor-in-Chief

PLOS Pathogens

orcid.org/0000-0002-7699-2064

Reviewer's Responses to Questions

**Part I - Summary**

Reviewer #1: In the manuscript “Early IL-17A production helps establish Mycobacterium intracellulare infection in mice”, Jung et al. provide a detailed characterization of a new mouse model of M. intracellulare infection. They examine numerous readouts of this novel infection model, including bacterial burdens in various organs, lung histology, cellular analysis, cytokine profiling, and time-to-death. The authors go on to examine the requirement of both neutrophils and IL-17A in the pathogenesis of this infection model, and they find that IL-17A levels, but not neutrophils, play a critical role in disease phenotypes.

This manuscript describes a very nice study with expansive data and strong conclusions. The multiple mouse infections, which each last for at least 39 weeks, represent an impressive research effort, and the amount of data mined from each infection and timepoint is admirable. The data throughout the manuscript are very clean and convincing—sometimes even quite striking (Fig 2A)—which results in an impressively thorough study with well-supported conclusions. And the inclusion of lung function tests was an especially nice touch. Moreover, I think examining two different manipulations of the immune response—neutrophil depletion and IL-17A blocking—and showing that one has an effect while another doesn’t is especially powerful; it’s strong experimental evidence that this infection model is not only useful for probing different facets of the immune response, but it also points to a specific role of IL-17A in the pathogenesis of NTM infection. Overall, this is an exciting new infection model that has a lot of potential for enabling many future discoveries about the pathogenesis of NTM infections.

The weaknesses I saw in this manuscript were the initial explanation/justification of how the model was developed and the thoroughness of the in vitro data presented in Figure 7; these are both detailed below.

Reviewer #2: Currently, animal models for NTM infection are imperfect. In this paper, Jung et al attempt to demonstrate a new model for pathologic MAI infection to suggest that IL-17 plays a pathogenic role in the progression of chronic NTM disease. use a high-dose intranasal inoculation f M. intracellulare into C57BL/6 mice. In this model, mice develop a progressive infection that leads to mortality a median of about 30 weeks after infection. Bacterial burden is progressive, and this leads to increased lung pathology over time. In this paper, the authors found that early expression of IL-17A contributes to pathology, while early depletion of neutrophils has little effect on outcomes. These observations are interesting and contribute to a field that needs better mouse models. Overall, I feel this paper is good, but I have some hesitation with its publication in its current form.

Reviewer #3: The study of Jung et al describes immunity in a chronic murine model of Mycobacterium intracellulare infection and the role of specific immune mediators in the control of infection. This is a somewhat comprehensive study that analyses a broad cross section of immune responses and provides a mechanistic role for IL-17 in M. intracellulare infection. The pathogenic role of IL-17 during infection with pathogenic mycobacteria is well established, but this study does provide some novel insights into the interplay between IL-17 and neutrophils during chronic mycobacterial infection.

**Part II – Major Issues: Key Experiments Required for Acceptance**

Reviewer #1: My primary experimental concern is with Figure 7; as the only in vitro data presented, I think it would benefit from a few additional conditions and/or readouts. For Fig 7A, it would be useful to see additional concentrations of IL-17A or different bacterial MOIs in order to see that more or less IL-17A/bacteria leads to higher or lower MMP-3 production. Looking at MMP-9 production in this set of experiments is also crucial, since it would strengthen the conclusion that MMP-3 is specifically regulated by IL-17A. In Fig 7B, it would be helpful to see CFUs at more than one timepoint to get a sense of the dynamics; are there differences in CFUs earlier during infection? Or later during infection? Are bacteria replicating in this cell line or being killed over time? Measuring cell death (LDH assay or something similar) might also strengthen this panel: if treatment leads to increased death or survival but there equal CFUs are recovered, there would be an overall difference in the number of bacteria per cell, which would be biologically meaningful.

My other major concern is understanding how the authors developed this infection model. The authors provide a nice description of similar mouse models and their shortcomings in the discussion, but were similar preliminary experiments done by the authors in the process of developing this model? Is there data that can be included from the authors trying other infection routes, other mouse strains, or different inoculums? Anything that would help establish how and why the authors chose these parameters for the rest of the paper. At the very least, a discussion of this decision-making process at the beginning of the results section would be beneficial.

I’m wondering why mice treated with anti-Ly6G weren’t fully analyzed (CFUs, histology, cytokines, cell analysis) at week 39 or 48. I’m especially curious about this since in mice treated with anti-IL-17A mAb, some key phenotypes like differences in CFUs and histology weren’t apparent until this late timepoint. If the late time point had been examined for the anti-Ly6G mice, would phenotypes be detectable?

For the histology images in S5 and S6, it would be helpful to have the sections blindly scored to provide unbiased, quantitative data. By eye, some lung sections look different but are described in the results as being similar (especially in S5, 2 wpi), so more representative images or unbiased scoring would strengthen/clarify this data.

Reviewer #2: Major concerns:

1. I think the paper's conclusions regarding the role of neutrophils in the pathogenesis of MAI infection in this model are somewhat confusing. They note in Figure 3B that neutrophils have a "double peak" with increases early in infection and late, but that depletion of neutrophils in the first four weeks of infection does not influence mortality. However, it seems plausible that ongoing neutrophil depletion might provide some protection if prolonged. The authors conclude in line 319 that "neutrophils are not essential to the establishment of MAI infeciton." However, that's not really what this paper is about -- the high dose of MAI is important for the establishment of infection. I think the conclusions must be tempered XXXX.

2. Models for NTM infection in small animals have not, to this point, been adequate surrogates for human disease. Furthermore, NTM disease is extremely diverse with skin, lung, bone, and disseminated infections and manifestations. It appears that this new model has characteristics of both fibronodular pulmonary infection ( with increased lung disease) as well as disseminated infection (due to high bacterial burden systemically. I am curious how the authors feel that this model fits into this field -- What type of disease is this model most like?

3.The rationale for choosing IL-17 as the treatment cytokine of choice based on data from Figure 3 and Figure S2 is not clear to me. Based on my reading of S2, it seems like many (32) cytokines were induced during the first 2 weeks of intranasal MAI infection. It might be useful to run a principal components analysis of all the cytokines in this data, then to define the groups of cytokines that are coexpressed with one another. I wouldn't be surprised some of the Th17 cytokines segregrated with one another. These might also provide clues toward cell specificity of cytokine responses when evaluated with flow cytometry cell analysis when available. Further, it will reduce the statistical burden of multiple comparisons in multiplexed cytokine data

Reviewer #3: 1. The examination of infection out to 39 weeks has revealed an unexpected imbalance in particular immune subsets, most strikingly a large accumulation of neutrophils at this late timepoint. Thus it was surprising that there was only a focus on neutrophils on the establishment of infection, and the impact of neutrophil depletion at late stages of infection was not examined. This would be important to determine if this late accumulation of neutrophils is a key driver of pathology/mortality in this model.

**Part III – Minor Issues: Editorial and Data Presentation Modifications**

Reviewer #1: If available, it would be nice to see images in Figure 1 or 2 of the whole lungs pre-sectioning to see if they had visible lesions like in lungs from Mtb-infected mice.

What was the endpoint used for time to death experiments? This should be included in the methods section.

The time to death data indicates that some mice die over the course of infection—some early and more later. Did these deaths also occur in the infection done to characterize each individual time point? There’s no indication that there were deaths in these groups (all groups are equal in size at n=6), so I’m wondering if extra mice were available/infected to account for loses/deaths along the way? It might be worth addressing this minor discrepancy somewhere in the text.

The panels from Fig 6G-H are duplicated in Fig S8.

I think labeling Fig 6C-M with the time point would help with clarity, especially because the schematic indicates multiple time points were assessed, but only one is presented in the main figure. Even though the info can be found in the legend, it’s nice to quickly see/understand that the data is from 39 weeks.

Line 253 mentions that MMP-9 levels are not affected by IL-17A blocking, but it doesn’t reference the MMP-9 data in Fig S9. The data in Fig S9E-F seem to show that MMP-9 does decrease—significantly so at 4 wpi—so a more accurate description might be that MMP-9 decreases transiently and modestly early in infection. I think this transient, modest decrease further stresses the importance of also examining MMP-9 levels in the in vitro infections in Figure 7.

The authors mention in the discussion the role of neutrophils during Toxoplasma and Listeria infection, but it would be useful to also mention the role of neutrophils during Mtb infection, especially since several other comparisons are drawn to Mtb throughout the discussion.

It would be interesting to know what aspects of human disease this model fails to recapitulate and maybe include in the discussion an analysis of what the limitations of this model are.

In lines 329-330, the authors mention that the clinical strain used in this model is resistant to a couple antibiotics, but it’s not clear how that is related to IL-17A or early responses; what is the biological relevance/importance of this antibiotic resistance in this context?

Reviewer #2: Minor issues:

1. Line 136 or so: It is not clear to me the numbers of mice in each experimental group in Figure 1. It appears 40 total mice were used, but was this 1:1 segregation 2:1? 3:1? Please clarify

2. Figure 7. It doesn't appear that MAI induces significant MMP3 above baseline in airway epithelial cells in your in vitro system. However, MAI induces pulmonary destruction during MAI infection. Does this observation suggest the possibility of an alternate mechanism for tissue destruction in your model of MAI infection?

3. There are multiple grammatical errors throughout the manuscript and this paper would benefit from a scientific editor.

4. There was no description in the methods of how CT scanning and lung physiological function studies were performed in the mouse, thus no way to assess if this work was done appropriately. Please add details to how these studies were conducted, the equipment used, and necessary details for these experiments at all areas in the paper where they were used.

5. Figure 4D. I would like more details as to the total numbers of CD45CD3- and CD45CD3+ cells producing IL-17. Although CD3+ cells produce significant IL-17, I would not be surprised if another cell type might have a greater frequency of cells that produce IL-17 (ILC3, for example). It seems a bit of a stretch to state that CD3+ cells are the primary cell type producing IL-17 when the flow panel is as sparse as it is

Reviewer #3: 2. In relation to the point above, the authors may wish to expand upon discussion of the potential role of KC during chronic infection, considering its association with neutrophil accumulation and heightened levels at 39 weeks post-infection.

3. The study required a very large dose of bacteria to achieve bacterial persistence and eventual mortality; the starting CFU seeded to the lung is >10e7 bacteria, and numbers increase by just over 10-fold after 39 weeks. Thus it is not necessarily true that this model ‘closely resembles the clinical presentation of human chronic NTM infection’ (see line 295); indeed the mortality incurred here could be to the unusually high dose delivered intranasally, which results in a much greater seeding of the lung compared to aerosol delivery (see reference 34 where aerosol delivery of a lower dose of bacteria resulted in greater initial replication and subsequent bacterial persistence). The authors may wish to better discuss the limitations of their model, considering the points raised above.

4. Interpretation of some of the data needs to be improved. The abstract (line 30) states that ‘depletion of neutrophils during the early 2 weeks of infection had no effect on bacterial burden, structure and function of lung and mortality until 48 weeks of infection’, however the data shows no difference at this late stage. Similarly, Line 368 states that ‘early IL-17A production helps M. intracellulare to establish infection in mouse lungs’, but the data shows that the impact of IL-17 depletion is most apparent at later stages of infection.

5. Bar graphs should show individual data points for each animal.

6. The study would benefit from a more through proof read to fix grammatical errors.

PLOS authors have the option to publish the peer review history of their article (what does this mean?). If published, this will include your full peer review and any attached files.

Reviewer #1: No

Reviewer #2: No

Reviewer #3: No
---

## [Decision Letter · Decision Letter 1]

17 Mar 2022

Dear Dr. Jung,

We are pleased to inform you that your manuscript 'Early IL-17A production helps establish Mycobacterium intracellulare infection in mice' has been provisionally accepted for publication in PLOS Pathogens.

Best regards,

Padmini Salgame

Associate Editor

PLOS Pathogens

JoAnne Flynn

Section Editor

PLOS Pathogens

Kasturi Haldar

Editor-in-Chief

PLOS Pathogens

orcid.org/0000-0001-5065-158X

Michael Malim

Editor-in-Chief

PLOS Pathogens

orcid.org/0000-0002-7699-2064

Reviewer Comments (if any, and for reference):

Reviewer's Responses to Questions

**Part I - Summary**

Reviewer #1: In the manuscript “Early IL-17A production helps establish Mycobacterium intracellulare infection in mice”, Jung et al. provide a characterization of a new mouse model for M. intracellulare infection, examining numerous readouts including bacterial burdens, histology, cellular analysis, cytokine profiling, and time-to-death. The authors also explore the requirement of neutrophils and IL-17A in the pathogenesis of this infection model and find that IL-17A levels, but not neutrophils, play a critical role in disease phenotypes.

My original concerns regarding how the infection model was established and the extent of the in vitro infection experiments were sufficiently addressed in the revised manuscript. My minor concerns were also sufficiently addressed.

Reviewer #2: The authors addressed our comments,nice work

Reviewer #3: This is a re-review of a previously submitted article. The authors have addressed the comments from the initial review.

**Part II – Major Issues: Key Experiments Required for Acceptance**

Reviewer #1: My previous concerns were sufficiently addressed by the authors, and I am satisfied with the revisions made to the manuscript.

Reviewer #2: (No Response)

Reviewer #3: (No Response)

**Part III – Minor Issues: Editorial and Data Presentation Modifications**

Reviewer #1: (No Response)

Reviewer #2: (No Response)

Reviewer #3: (No Response)

PLOS authors have the option to publish the peer review history of their article (what does this mean?). If published, this will include your full peer review and any attached files.

Reviewer #1: No

Reviewer #2: No

Reviewer #3: No

---

## [Editor Report · Acceptance letter]

29 Mar 2022

Dear Dr. Jung,

We are delighted to inform you that your manuscript, "Early IL-17A production helps establish Mycobacterium intracellulare infection in mice," has been formally accepted for publication in PLOS Pathogens.

Best regards,

Kasturi Haldar

Editor-in-Chief

PLOS Pathogens

orcid.org/0000-0001-5065-158X

Michael Malim

Editor-in-Chief

PLOS Pathogens

orcid.org/0000-0002-7699-2064